# Impact of early pregnancy body mass index and gestational weight gain on birth outcomes: Findings from a pregnancy cohort in South Delhi, India

Saijuddin Shaikh<sup>☯</sup>, Ranadip Chowdhury<sup>☯</sup>*, Neeta Dhabhai<sup>‡</sup>, Sunita Taneja<sup>☯</sup>, Nita Bhandari<sup>☯</sup>

Society for Applied Studies, New Delhi, India

☯ These authors contributed equally to this work.
‡ ND also contributed equally to this work.
* ranadip.chowdhury@sas.org.in

## Abstract

Early pregnancy low body mass index (BMI) and inadequate gestational weight gain (IGWG) are significant risk factors for adverse birth outcomes. However, the specific risks among underweight women with IGWG or excessive GWG (EGWG), as well as overweight/obese women with IGWG or EGWG, compared to normal BMI women with adequate GWG (AGWG), remain poorly defined. The primary objective is to estimate the risk of small for gestational age (SGA) among women with underweight at early pregnancy and IGWG compared to women with normal weight at early pregnancy and AGWG. Data were derived from a randomized factorial trial. BMI was assessed at <14 weeks of gestation, and women weight was measured monthly until 32 weeks, biweekly until 36 weeks and weekly until delivery. GWG was classified as per Institute of Medicine (IOM) guidelines. Infant weight and length were measured <14 days of birth. Regression analysis assessed associations (risk ratio, RR) between BMI, GWG, and birth outcomes. The highest prevalence of SGA (62.4%; 95%CI 57.1–67.5) was observed among underweight women with IGWG. Normal BMI women with IGWG (aRR 1.36; 95%CI 1.21–1.53) and underweight women with IGWG (aRR 1.65; 95%CI 1.44–1.89) increased, and overweight/obese women with excessive GWG (EGWG) reduced risk of SGA (aRR 0.56; 95%CI 0.43–0.72), compared to normal BMI women with AGWG. IGWG among normal women (aRR 1.43; 95%CI 1.21–1.70) and IGWG among underweight women (aRR 2·09; 95%CI 1·74–2·52) also increased, and overweight/obese women with EGWG (aRR 0.73; 95%CI 0.54–0.98) reduced the risk of low birth weight (LBW). Underweight women with IGWG face the highest risk of adverse outcomes, while EGWG among overweight or obese women reduces the risk of adverse outcomes. Monitoring of GWG alongside early pregnancy BMI is essential for guiding targeted nutritional intervention to improve pregnancy outcomes.

**Data availability statement:** Please find the dataset DOI: 10.5061/dryad.6t1g1jxbs.

**Funding:** The study was funded by Biotechnology Industry Research Assistance Council (BIRAC) of the Department of Biotechnology, Government of India (GCI-ACT ref No BIRAC/GCI/0085/03/14-ACT to NB) and by the Bill & Melinda Gates Foundation, USA (grant ID OPP1191052 to NB). The funders had no role in study design, data collection and analysis, decision to publish, or preparation of the manuscript.

**Competing interests:** The authors have declared that no competing interests exist.

## Introduction

Small and vulnerable newborns are significant public health challenges. Term SGA babies are heavily concentrated in South Asia, accounting for nearly two-thirds (63%) of the global burden [1]. Additionally, South Asia has the highest prevalence of preterm birth at 13.2% [1,2]. In India, the prevalence of preterm, term SGA, LBW and stunting are comparatively higher than other low and middle income countries (LMICs) [3,4]. These vulnerable babies are associated with higher risks of morbidity, mortality and growth faltering during infancy and childhood, poor cognitive performance, and developmental delays compared to their healthy peers. These children are also decreased opportunities in education and employment, and a higher likelihood of non-communicable diseases later in life compared to normal infants [5–8].

Gestational weight gain (GWG) plays a pivotal role in determining pregnancy outcomes and long-term health trajectories for both mother and child [9]. Inadequate GWG is a well-established risk factor for adverse outcomes, including LBW, SGA infants, and preterm birth [10,11]. On the other hand, excessive GWG has been associated with higher risk of cesarean section, large for gestational age, macrosomia, maternal weight retention and obesity in the offspring later in life [11–13]. Prepregnancy nutritional status is another vital determinant risk factor of birth outcomes. Evidence suggested that lower prepregnancy BMI is associated with preterm birth, while higher BMI is linked to large for gestational age and macrosomia [14,15]. However, the combined effect of early pregnancy BMI and trimester-specific GWG on birth outcomes in LMICs is scared. A study in India has found the higher risk of adverse birth outcomes among women whom GWG was lower than the IOM guidelines compared to those who had within the IOM guidelines [16]. That study did not mention the effect of total and trimester-specific GWG for underweight or overweight/obese women on adverse birth outcomes. The US Institute of Medicine guidelines 2009 provide recommendations for GWG based on prepregnancy BMI [17], however, there is a lack of GWG guidelines designed to be relevant to populations in LMICs. Therefore, the extent to which these risks vary across BMI and GWG categories, particularly in South Asia, remains unclear.

India is the country where underweight and overweight are the public health concern, and it is important to clarify the combined effect of early pregnancy BMI and trimester-specific GWG on the risk of adverse birth outcomes. We hypothesize that early pregnancy underweight (BMI < 18.5 kg/m$^2$) women with inadequate gestational weight gain increase the risk of small for gestational age babies, compared to normal body mass index women with adequate gestational weight gain. We also aimed to explore the association of gestational weight gain and early pregnancy body mass index with low birth weight, preterm, stunting and wasting.

## Materials and methods

### Ethics statement

The Ethics Review Committee of the Society for Applied Studies, New Delhi (SAS/ERC/LG/2027), Vardhman Medical College, Safdarjung Hospital, New Delhi (IEC/

SJH/VMMC/PROJECT-2017/694) and the World Health Organization, Geneva (ERC.0002934) have approved the study. The study was conducted in accordance with the ethical principles outlined in the Declaration of Helsinki. The trial was registered with the Clinical Trials Registry of India (CTRI/2017/06/008908). Written informed consent was obtained from all mothers prior to participation. All data were collected in a private area of the participants' homes, and confidentiality of the collected data was maintained.

## Study design and data source

This analysis used data from Women and Infants Integrated Intervention for Growth Study (WINGS), an unmasked factorial individually randomized controlled trial conducted in urban and peri-urban low-to-middle income communities in South Delhi, India. Details about the main trial (WINGS) have been published elsewhere [2]. Prepregnant women were enrolled from 1 July 2017–30 December 2019 and were followed for the period of 18 months, from 01 January 2020–30 June 2021.

Briefly, a household survey was conducted to identify eligible married women between 18–30 years of age, living with husband, and staying permanently in the study area. After a written informed consent (first consent), height (Seca-213 stadiometer) and weight (Salter 9509 weighing scale) were measured and women were randomized (first randomization) into a preconception intervention and control groups. All women were followed up till they became pregnant or 18 months from enrolment. If any woman did not become pregnant within 18 months from enrolment, the woman was excluded from the study. An ultrasound (GE ultrasound Voluson S8, PI Healthcare Inc., 23865 Via Del Rio, Yorba Linda CA 92887, USA) was done between 9–13 weeks to confirm pregnancy for women who reported pregnancy [2]. Within 14 weeks of gestation, eligible participants were rescreened and written informed consent (second consent) obtained again for further participation in the study and randomized again (second randomization) to either receive interventions or routine care in control group.

Four intervention groups were a) preconception, pregnancy and childhood, b) preconception, c) pregnancy and early childhood, and d) routine care. Details regarding interventions and interventions delivery have been discussed previously [2]. Briefly, interventions were given during pre- and peri-conception on health, nutrition, psychosocial care and support, and water, sanitation and hygiene. Pregnant women in intervention group with BMI between 16 kg/m² and 18.5 kg/m² were provided locally made snacks containing 500 kcal energy and 6–10 g protein per packet, whereas double amount was given to the women with BMI < 16 kg/m². As a source of high quality protein, all women with BMI < 21 kg/m² were supplemented one egg or 180 ml milk (70 kcal energy and 6 g protein) six days a week. All pregnant women were given one recommended daily allowance of micronutrients supplement daily during the pregnancy period. On the other hand, pregnant women in control group were advised to seek routine care from government sources or private health providers. In intervention group, study team visited participants' home weekly to reinforce interventions, replenish supplies and record compliance. During pregnancy, weight was assessed monthly at home or at the study clinic until 32 weeks, biweekly until 36 weeks and weekly until delivery to monitor GWG. A study team trained in anthropometry measured infant weight and length <14 days of birth. Anthropometry of 52% neonates was measured within 72 hours of birth and 93% of infants within 7 days of life.

The family wealth index was calculated for each participant by performing a principal component analysis based on all 33 assets owned by the household as done in national surveys [18]. The total scores were used to divide the population into five equal wealth quintiles: the poorest, very poor, poor, less poor and least poor.

## Enrolment of the subjects in the current study

Women from the previous study [2] who became pregnant within 18 months of enrolment were included in the present study. Early pregnancy BMI was calculated and women were classified as underweight, normal weight, overweight or obese. Eligible women were then randomised to receive either the intervention or routine antenatal care.

## Variables

**Definition of primary exposure variables.** Maternal early pregnancy BMI and GWG were exposure variables. BMI at enrolment (<14 weeks of gestation) was calculated using this formula; weight (kg) divided by height (m) square. BMI was categorized to define underweight (BMI < 18.5 kg/m²), normal (BMI 18.5–<25 kg/m²) and overweight or obese (BMI ≥ 25 kg/m²). GWG at different time points was calculated in the following way. Total GWG (kg) from enrolment to last visit of before delivery: the weight recorded at enrolment was subtracted from the last recorded weight before delivery. GWG between enrolment and 26 weeks: the weight measured at enrolment was subtracted from weight measured at 26 weeks. Weight gain from 27 weeks to last visit of before delivery: the weight recorded at 27 weeks was subtracted from the last recorded weight before delivery. GWG was categorized in to inadequate, adequate and excessive GWG using the IOM guideline [17] and calculation was reported elsewhere [16]. Inadequate GWG: < 0.44 kg/week for BMI < 18.5 kg/m², < 0.35 kg/week for BMI 18.5–24.99 kg/m², < 0.23 kg/week for BMI 25.0–29.99 kg/m², and <0.17 kg/week for BMI ≥ 30 kg/m². Adequate GWG: 0.44 to 0.58 kg/week for BMI < 18.5 kg/m², 0.35 to 0.50 kg/week for BMI 18.5–24.99 kg/m², 0.23 to 0.33 kg/week for BMI 25.0–29.99 kg/m², and 0.17 to 0.27 kg/week for BMI ≥ 30 kg/m². Excessive GWG: > 0.58 kg/week for BMI < 18.5 kg/m², > 0.50 kg/week for BMI 18.5–24.99 kg/m², > 0.33 kg/week for BMI 25.0–29.99 kg/m², and >0.27 kg/week for BMI ≥ 30 kg/m². A nine groups categorical variable was generated by combining early pregnancy BMI and GWG categories, and normal weight with AGWG was used as the comparator. Also a single variable of four intervention groups was created.

**Definition of outcome variables.** Continuous outcome variables were weight and length. Categorical outcome variables were term SGA (<10th percentile of a INTERGROWTH-21 standards by gestational age and sex) [19], preterm (<37 weeks of gestational age), stunting and wasting. Stunting and wasting were defined as length-for-age and weight-for-length z scores < -2SD respectively according to World Health Organization growth standards [20]. Weight less than 2500 g measured <14 days of birth was considered to define LBW. Gestational age in weeks at birth was calculated by estimating the duration in weeks from the date of the ultrasound examination to the date of birth, and adding this to the gestational age in weeks at the time of the ultrasound, as determined using the INTERGROWTH-21 standards [21].

**Inclusion criteria.** The following inclusion criteria were set for this analysis: antenatal women with gestational age < 14 weeks at enrolment, singleton pregnancy and neonates delivered by these women within <14 days of birth.

**Exclusion criteria.** Women living in temporary housing and those moving away from study area were excluded.

**Power calculation.** The available sample size had at least 80% power to detect a RR of 1.4 between BMI 18.5 and 24.99 kg/m² and IGWG and adverse pregnancy outcomes, assuming an adverse birth outcome prevalence between 7% and 35% among women with BMI 18.5 and 24.99 kg/m² and AGWG with 5% significance level.

## Statistical analyses

Continuous sociodemographic variables were reported as mean (SD) given their normal distribution, while categorical variables were summarized as frequency (%). Based on previous evidences, a Directed Acyclic Graph (http://www.dagitty.net/) was developed to show the potential confounders by visually representing the casual relationships between variables in the study (**S1 Fig**). We included variables that changed risk ratio (RR) and β-coefficient of the outcome variables by 5%–10% in the univariable models included in the multivariate models; these were maternal age, maternal education, intervention and family wealth quintile. For binary outcomes, including term SGA, LBW, preterm birth, stunting, and wasting, we fitted generalized linear models (GLMs) with a log link to estimate risk ratios (RRs) and 95% confidence intervals (CIs). Models of the binomial family were applied where possible, with robust Poisson models used in instances of non-convergence [22]. For continuous outcomes such as infant birthweight and length, we used GLMs from the Gaussian family with an identity link to estimate mean differences and 95% CIs. Each analysis followed a two-step approach: first presenting unadjusted associations and subsequently adjusting for pre-specified covariates, with both sets of estimates displayed in the tables. Results for total GWG were summarized in forest plots, while trimester-specific GWG findings were presented in tabular form. To assess potential effect modification, we generated regression lines with 95% CIs

illustrating the interaction between early-pregnancy BMI and GWG. In addition, we calculated and depicted the prevalence of term SGA, LBW, stunting, and preterm birth across categories of maternal BMI and GWG using bar diagrams. All analyses were performed using Stata version 14 (StataCorp, College Station, TX, USA). Statistical significance was defined as $p < 0.05$, with 95%CI.

## Results

At second randomization, 4921 pregnant women whose weight and height were measured during prepregnancy were enrolled in this study. Based on the inclusion and exclusion criteria, baseline information was available for 3855 women, and anthropometric measurements were collected for 3695 singleton infants (<14 days of birth), who were included in the analysis (**Fig 1**). [**Fig 1**. *Flow diagram of enrolment, randomization, followup and outcomes*].

Sociodemographic characteristics of the participated women are presented in **Table 1**.

### Association of early pregnancy BMI and GWG with infant weight and length

Linear regression analysis was done to find the associations between early pregnancy BMI, GWG, and continuous outcomes (infant weight in g and length in cm) and unadjusted and adjusted β-values are shown in S1 Table and S2 Table. For total GWG, IGWG among normal BMI women (adjusted β -126·62; 95%CI -163·78– -89·46) and IGWG among underweight women (adjusted β -239·12; 95%CI -289·53– -188·72) were associated with LBW compared to normal BMI women with AGWG. Conversely, EGWG among normal BMI (adjusted β 92·22; 95%CI 40·39–144·04) and overweight or obese women (adjusted β 116·26; 95%CI 65·38–167·14) was associated with higher birth weight. Similar trends were observed for length, with stronger associations in the second trimester compared to the third trimester.

### Distribution of adverse birth outcomes by early pregnancy BMI and GWG

The overall prevalence of term SGA (62.4%; 95%CI 57.1–67.5), LBW (47.3%; 95%CI 42.0–52.7), stunting (37.3%; 95%CI 32.3–42.6) and preterm (10.1%; 95%CI 7.3–13.8) among infants of underweight women (BMI < 18·5 kg/m²) with IGWG were higher compared to other groups (**S2 Fig**). Same pattern was observed for trimester-specific GWG (**S3** and **S4 Figs**).

### Association between early pregnancy BMI, GWG, and term SGA

For total GWG, after adjusting for confounders, IGWG among normal BMI women (aRR 1.36; 95%CI 1.21–1.53) and IGWG among underweight women (aRR 1.65; 95%CI 1.44–1.89) were found to be higher risk of term SGA, and overweight or obese with EGWG (aRR 0.56; 95%CI 0.43–0.72) was associated with reduced the risk of SGA (**Fig 2A**). In the second trimester, IGWG among underweight women (aRR 1.44; 95%CI 1.27–1.64) was found to be a higher risk. In contrast, normal BMI women with EGWG (aRR 0.75; 95%CI 0.62–0.91), overweight or obese with AGWG (aRR 0.55; 95%CI 0.41–0.73), overweight or obese with IGWG (aRR 0.84; 95%CI 0.71–0.99) and overweight or obese with EGWG (aRR 0.50; 95%CI 0.39–0.65) were associated with reduced the risk of SGA. Similar trends were observed in the third trimester (**Table 2**). [**Fig 2A**. *Associations of early pregnancy BMI and total GWG with term SGA*].

### Association between early pregnancy BMI, GWG, and LBW

Adjusted associations between early pregnancy BMI and total GWG with LBW is shown in **Fig 2B**. While, unadjusted and adjusted associations in the second and third trimesters are shown in **Table 2**. Inadequate GWG among normal BMI women (aRR 1·43; 95%CI 1·21–1·70) and IGWG among underweight women (aRR 2·09; 95%CI 1·74–2·52) were associated with an increased risk of LBW, compared to normal BMI women with AGWG. Conversely,

EGWG among overweight or obese women was associated with a reduced risk of LBW (aRR 0·73; 95%CI 0·54–0·98) (**Fig 2B**). Similar trends were observed for increasing the risk in the second trimester. Though, normal BMI with EGWG (aRR 0.73; 95%CI 0.56–0.95) and AGWG among overweight or obese women (aRR 0·66; 95%CI 0·47–0·93) reduced the

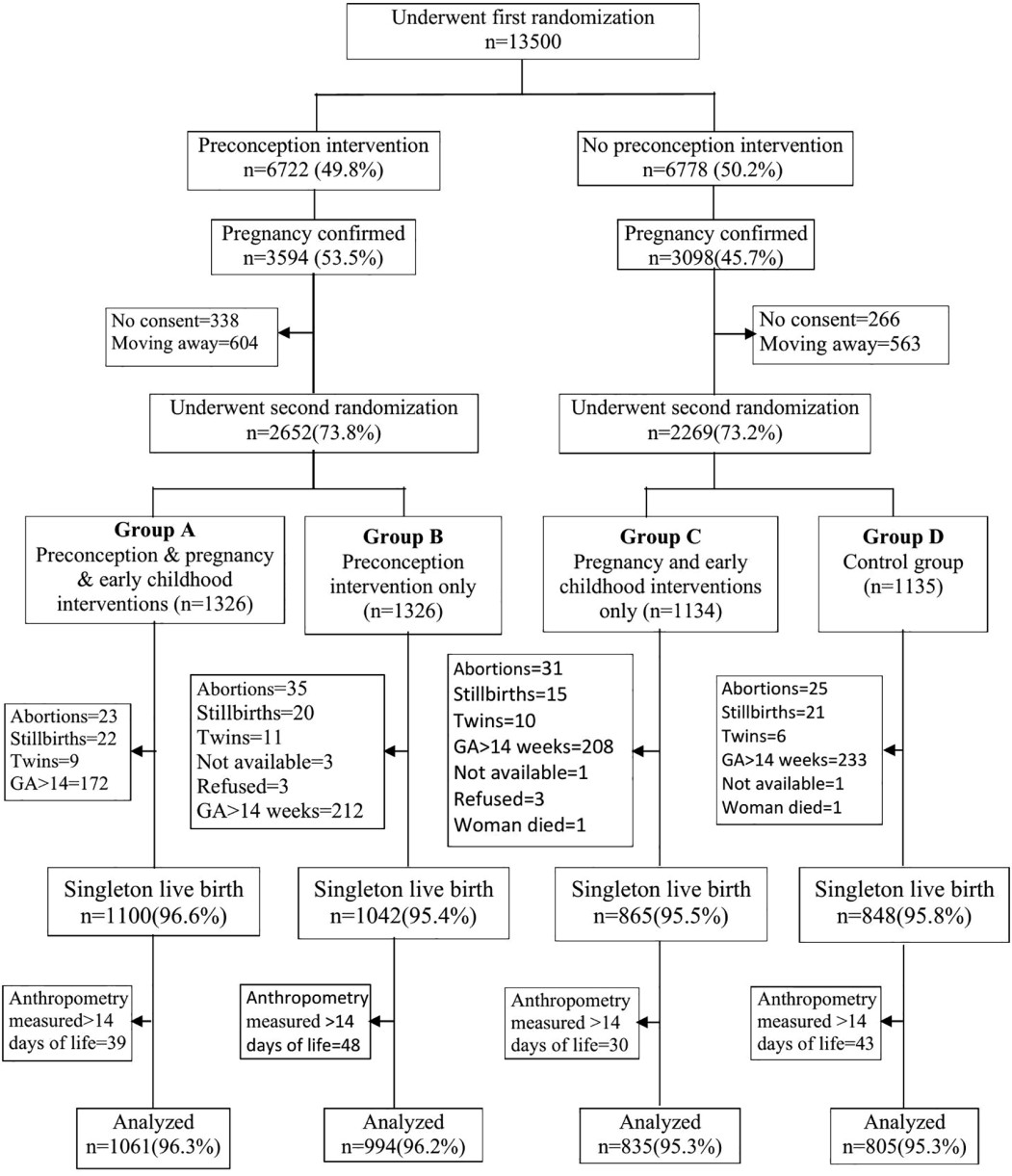

**Fig 1. GA, gestational age.**

risk of LBW, but no protective effect of AGWG or EGWG was observed in the third trimester (**Table 2**). [**Fig 2B**. *Associations of early pregnancy BMI and total GWG with LBW*].

## Association between early pregnancy BMI, GWG, and preterm

After adjusting for maternal age, maternal education, family wealth quintile, and intervention effect, RRs for total GWG in preterm birth are shown in **Fig 2C**. EGWG among normal BMI women was associated with a reduced risk of preterm birth (aRR 0·44; 95%CI 0·22–0·88), compared to normal BMI women with AGWG. In the second trimester, normal BMI women with IGWG (aRR 1.38; 95%CI 1.03–1.86), underweight women with IGWG (aRR 1.65; 95%CI 1.15–2.37), overweight or

**Table 1. Sociodemographic characteristics of the study participants at enrolment[a].**

| Characteristics | n=3855 |
|---|---|
| Family having below poverty line card, n (%) | 153 (4·0) |
| Family covered by health insurance scheme, n (%) | 418 (10·8) |
| Age at enrolment (years), mean (SD) | 23·8 (3·0) |
| Weight at enrolment (kg), mean (SD) | 52.3 (10.0) |
| Height at enrolment (cm), mean (SD) | 152·4 (5·6) |
| Early pregnancy body index (kg/m²), n (%) | |
| Underweight (<18·5 | 586 (15·2) |
| Normal weight (18·5 – 24·9) | 2311 (60·0) |
| Overweight (25 – 29·9) | 792 (20·5) |
| Obese (≥30) | 166 (4·3) |
| Gestational age at enrolment (weeks), mean (SD) | 10·9 (1·4) |
| Gestational age at delivery (weeks), mean (SD) | 38·7 (1·5) |
| Total gestational weight gain (kg), mean (SD) | 8.2 (3.6) |
| Parity, n (%) | |
| 0 | 1633 (42·4) |
| 1 | 2222 (57·6) |
| Year of schooling, mean (SD) | 10·5 (4·1) |
| Women occupation, housewife, n (%) | 3668 (95·2) |
| Family type (joint), n (%) | 2172 (56·3) |
| Wealth quintile, n (%) | |
| Poorest | 593 (15·4) |
| Very poor | 767 (19·9) |
| Poor | 818 (21·2) |
| Less poor | 875 (22·7) |
| Least poor | 802 (20·8) |

[a]Data presented as mean (SD) or frequency (%).

obese women with IGWG (aRR 1.48; 95%CI 1.03–2.12), and overweight or obese women with EGWG (aRR 1.64; 95%CI 1.12–2.42) increased the risk of preterm, compared to normal BMI women with AGWG (**Table 3**). Similar pattern was observed in the third trimester. However, degree of risk was higher the third trimester compared to the second trimester (**Table 3**). [Fig 2C. *Associations of early pregnancy BMI and total GWG with preterm*].

## Association between early pregnancy BMI, GWG and stunting and wasting

**Fig 3A** and **3B** show the adjusted associations of early pregnancy BMI and total GWG with stunting and wasting. Total IGWG among normal BMI women was associated with increased risk of stunting (aRR 1·44; 95%CI 1·21–1·70), and wasting (aRR 1·36; 95%CI 1·09–1·71). Among underweight women with IGWG, the risks of stunting (aRR 1·78; 95%CI 1·46–2·17), and wasting (aRR 1·58; 95%CI 1·20–2·10) were significantly higher. However, EGWG among overweight or obese women was protective effect for wasting (aRR 0·58; 95%CI 0·38–0·87). No significant association was found of overweight or obese with EGWG

with stunting (**Fig 3A**). Similar patterns were observed for trimester-specific GWG among BMI category, though associations were stronger in the third trimester compared to the second trimester (Tables 3 and 4). [Fig 3A. *Associations of early pregnancy BMI and total GWG with stunting*; Fig 3B. *Associations of early pregnancy BMI and total GWG with wasting*]

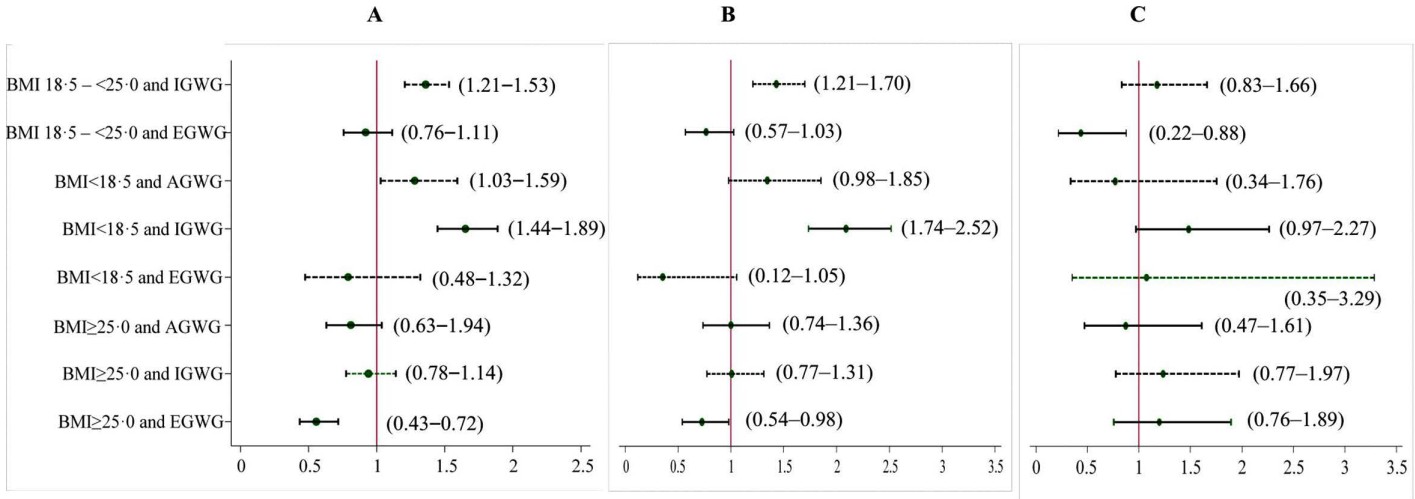

**Fig 2. GLMs of the Binomial or Poisson family with a log-link function was used to calculate RR after adjusting for maternal age, maternal education, family wealth quintile and intervention.** Early pregnancy nutritional status was defined using the following cut-off pints: underweight (BMI < 18.5), normal weight (BMI 18.5–24.9), and overweight or obese (BMI ≥ 25). GWG was categorized as inadequate, adequate and excessive GWG according to the IOM cut-off points. Normal weight with AGWG were used as comparator to calculated RR and presented on X-axis with 95%CI. AGWG, adequate gestational weight gain; BMI, body mass index; CI, confidence intervals; EGWG, excessive gestational weight gain; GLM, generalized linear model; GWG, gestational weight gain; IGWG, inadequate gestational weight gain; IOM, Institute of medicine; LBW, low birth weight; RR, risk ratio; SGA, small for gestational age.

## Interaction between early pregnancy BMI and GWG

Early pregnancy BMI significantly influenced the relationship between total GWG and infant weight and length. Notable interactions were observed for infant weight (p = 0·008) (**Fig 4A**) and length (p = 0·036) (**Fig 4B**). The association between GWG (kg) per week and these outcomes was more pronounced among underweight women (BMI < 18·5 kg/m²) compared to overweight or obese women (BMI ≥ 25 kg/m²) respectively. [**Fig 4A**. *Relationship between GWG during pregnancy and infant weight stratified by early pregnancy BMI;* **Fig 4B**. *Relationship between GWG during pregnancy and infant length stratified by early pregnancy BMI*]

## Discussion

Approximately 15% of women in the study were underweight, while 25% were overweight or obese. Among the participants, 10% were underweight with IGWG, 3.4% were underweight with AGWG, and 21.8% had a normal BMI with AGWG. The prevalence of term SGA (62.4%), LBW (47.3%), stunting (37.3%), and preterm birth (10%) were higher among underweight women with IGWG compared to other groups. The risk of LBW was two times, term SGA 1.65 times, stunting 1.78 times, and wasting 1.58 times higher among underweight women with IGWG compared to women with normal BMI and AGWG. Conversely, among overweight or obese women with EGWG, the risk of LBW was reduced by 27% and followed by SGA 44%, and wasting 42% compared to women with normal BMI and AGWG.

Research from other low- and middle-income countries [23] has reported that EGWG among women with higher early pregnancy BMI is related with increased birth weight, while IGWG is associated with LBW. Similarly, studies from sub-Saharan Africa have reported that IGWG increased and EGWG reduced the risk of LBW. These findings highlight the importance of AGWG to reduce the risk of LBW, particularly among underweight women [24,25].

Underweight women with IGWG were more likely to deliver preterm birth, with a higher risk occurring during the third trimester compared to the second trimester. The likelihood of having preterm birth was different across different BMI

**Table 2. Association between early pregnancy BMI and GWG with SGA and LBW during second and third trimesters.**

| Variables | SGA (RR 95% CI) | | | | LBW (RR 95% CI) | | | |
|---|---|---|---|---|---|---|---|---|
| | Unadjusted | P value | Adjusted[a] | P value | Unadjusted | P value | Adjusted[a] | P value |
| *GWG from enrolment to 26 weeks of gestation (n = 3502)* | | | | | | | | |
| BMI 18·5 –<25·0 and AGWG | Ref | | Ref | | Ref | | Ref | |
| BMI 18·5 –<25·0 and IGWG | 1·13(1.01–1·27) | 0.034 | 1.11(0.99–1·24) | 0.076 | 1·23(1·05–1·43) | 0.010 | 1·21(1·03–1·41) | 0.017 |
| BMI 18·5 –<25·0 and EGWG | 0·74(0·61–0·90) | 0.003 | 0.75(0.62–0·91) | 0.004 | 0·73(0·56–0·95) | 0.019 | 0·73(0·56–0·95) | 0.021 |
| BMI<18·5 and AGWG | 1·24(1·02–1·50) | 0.028 | 1.20(0.99–1·45) | 0.069 | 1·23(0·93–1·62) | 0.146 | 1·20(0·91–1·58) | 0.198 |
| BMI<18·5 and IGWG | 1·50(1·32–1·71) | 0.000 | 1.44(1.27–1·64) | 0.000 | 1·93(1·63–2·29) | 0.000 | 1·87(1·57–2·22) | 0.000 |
| BMI<18·5 and EGWG | 0·93(0·64–1·34) | 0.686 | 0.91(0.63–1·31) | 0.616 | 0·76(0·43–1·34) | 0.340 | 0·75(0·43–1·34) | 0.333 |
| BMI≥25·0 and AGWG | 0·54(0·41, 0·72) | 0.000 | 0.55(0.41–0·73) | 0.000 | 0·66(0·47–0·92) | 0.014 | 0·66(0·47–0·93) | 0.017 |
| BMI≥25·0 and IGWG | 0·84(0·71–0·99) | 0.043 | 0.84(0.71–0·99) | 0.046 | 1·03(0·83–1·27) | 0.820 | 1·02(0·82–1·26) | 0.861 |
| BMI≥25·0 and EGWG | 0·49(0·38–0·63) | 0.000 | 0.50(0.39–0·65) | 0.000 | 0·74(0·57–0·98) | 0.033 | 0·76(0·58–1·00) | 0.051 |
| *GWG from 27 weeks to the last visit of before delivery (n = 2949)* | | | | | | | | |
| BMI 18·5 –<25·0 and AGWG | Ref | | Ref | | Ref | | Ref | |
| BMI 18·5 –<25·0 and IGWG | 1·22(1·07–1·40) | 0.004 | 1.22(1.07–1·40) | 0.003 | 1·38(1·13–1·68) | 0.002 | 1.38(1.13–1·68) | 0.002 |
| BMI 18·5 –<25·0 and EGWG | 0·98(0·84–1·15) | 0.806 | 0.98(0.84–1·45) | 0.824 | 1·05(0·83–1·32) | 0.685 | 1.05(0.83–1·32) | 0.684 |
| BMI<18·5 and AGWG | 1·13(0·86–1·48) | 0.378 | 1.10(0.84–1·44) | 0.495 | 1·62(1·16–2·26) | 0.005 | 1.60(1.14–2·24) | 0.006 |
| BMI<18·5 and IGWG | 1·64(1·42–1·90) | 0.000 | 1.58(1.37–1·83) | 0.000 | 2·17(1·76–2·67) | 0.000 | 2.13(1.72–2·63) | 0.000 |
| BMI<18·5 and EGWG | 1·15(0·83–1·60) | 0.399 | 1.11(0.80–1·54) | 0.537 | 1·15(0·70–1·90) | 0.579 | 1.13(0.68–1·86) | 0.643 |
| BMI≥25·0 and AGWG | 0·76(0·57–1·01) | 0.062 | 0.79(0.59–1·05) | 0.110 | 0·86(0·58–1·28) | 0.462 | 0.88(0.59–1·30) | 0.511 |
| BMI≥25·0 and IGWG | 0·73(0·55–0·97) | 0.030 | 0.75(0.57–0·99) | 0.049 | 0·90(0·62–1·29) | 0.559 | 0.91(0.63–1·31) | 0.604 |
| BMI≥25·0 and EGWG | 0·62(0·50–0·76) | 0.000 | 0.64(0.52–0·79) | 0.000 | 0·85(0·66–1·11) | 0.231 | 0.86(0.67–1·12) | 0.276 |

Values are calculated using GLMs of the Binomial or Poisson family with a log-link function, compared with reference value, normal BMI with AGWG, and presented as RR and 95% confidence intervals, and p value.

AGWG, adequate gestational weight gain; BMI, body mass index; CI, confidence intervals; EGWG, excessive gestational weight gain; IGWG, inadequate gestational weight gain; LBW, low birth weight; RR, risk ratio; SGA, small for gestational age.

[a]Adjusted variables: Maternal age, maternal education, family wealth quintile and intervention.

categories and GWG levels. A greater risk was found among women who were already overweight or obese and also gained an excessive weight during pregnancy. These findings are consistent with study from other countries [26–28]. They have shown a higher chance of preterm birth in pregnant women with EGWG, specially during the third trimester. In the analysis, a positive association was found between SGA and IGWG among underweight and normal weight women, while EGWG among overweight or obese women reduced the SGA. The degree of association varied by trimester, with stronger effects observed in the second trimester. These findings are consistent with previous studies, including a meta-analysis by Goldstein et al. (2018) [13]. They concluded that IGWG is a risk of SGA, particularly among underweight women. Another study has reported the similar finding that IGWG increases and AGWG in the second trimester reduces the risk of SGA [26]. These results underscore the importance of AGWG particularly during the second trimester, to reduce the risk of SGA. The combined effects of early pregnancy BMI and GWG were also evident for stunting and wasting. Risks of these outcomes increased among underweight and normal weight women with IGWG, and stronger effects observed among underweight women. Limited studies have explored these associations, findings from rural Malawi and Vietnam supported the role of AGWG in reducing stunting [24,29]. These results highlight the need for targeted interventions to improve GWG among underweight women to optimize infant growth outcomes.

A systematic review conducted in 25 LMICs reported that 54% and 22% of women had inadequate and excessive GWG, respectively [30]. Although the prevalence of IGWG in our study was lower than that reported in the review, the

**Table 3. Association between early pregnancy BMI and GWG with preterm and stunting during second and third trimesters.**

| Variables | Preterm (RR 95% CI) | | | | Stunting (RR 95% CI) | | | |
|---|---|---|---|---|---|---|---|---|
| | Unadjusted | P value | Adjusted[a] | P value | Unadjusted | P value | Adjusted[a] | P value |
| *GWG from enrolment to 26 weeks of gestation (n = 3502)* | | | | | | | | |
| BMI 18·5 −<25·0 and AGWG | Ref | | Ref | | Ref | | Ref | |
| BMI 18·5 −<25·0 and IGWG | 1·43(1·06–1·91) | 0.017 | 1·38(1·03–1·86) | 0.030 | 1.29(1.10–1.52) | 0.003 | 1.29(1.10–1.52) | 0.002 |
| BMI 18·5 −<25·0 and EGWG | 0·74(0·46–1·21) | 0.236 | 0·75(0·46–1·22) | 0.250 | 0.71(0.54–0.94) | 0.019 | 0.71(0.54–0.94) | 0.017 |
| BMI < 18·5 and AGWG | 1·24(0·73–2·13) | 0.430 | 1·21(0·70–2·07) | 0.497 | 0.96(0.69–1.34) | 0.812 | 0.93(0.66–1.31) | 0.692 |
| BMI < 18·5 and IGWG | 1·71(1·19–2·45) | 0.003 | 1·65(1·15–2·37) | 0.006 | 1.86(1.55–2.23) | 0.000 | 1.82(1.51–2.18) | 0.000 |
| BMI < 18·5 and EGWG | 0·90(0·34–2·37) | 0.828 | 0·91(0·35–2·42) | 0.856 | 0.73(0.40–1.35) | 0.316 | 0.73(0.40–1.33) | 0.300 |
| BMI ≥ 25·0 and AGWG | 1·42(0·90–2·22) | 0.129 | 1·39(0·88–2·28) | 0.156 | 0.79(0.57–1.09) | 0.151 | 0.82(0.60–1.13) | 0.229 |
| BMI ≥ 25·0 and IGWG | 1·52(1·06–2·17) | 0.022 | 1·48(1·03–2·12) | 0.035 | 1.01(0.81–1.27) | 0.905 | 1.04(0.83–1.31) | 0.714 |
| BMI ≥ 25·0 and EGWG | 1·63(1·12–2·38) | 0.011 | 1·64(1·12–2·42) | 0.011 | 0.84(0.64–1.10) | 0.214 | 0.87(0.66–1.15) | 0.331 |
| *GWG from 27 weeks to the last visit of before delivery (n = 2949)* | | | | | | | | |
| BMI 18·5 −<25·0 and AGWG | Ref | | Ref | | Ref | | Ref | |
| BMI 18·5 −<25·0 and IGWG | 2·37(1·48–3·80) | 0.000 | 2·40(1·50–3·84) | 0.000 | 1.31(1.07–1.60) | 0.008 | 1.34(1.10–1.64) | 0.004 |
| BMI 18·5 −<25·0 and EGWG | 1·24(0·71–2·16) | 0.458 | 1·23(0·70–2·15) | 0.468 | 0.96(0.76–1.21) | 0.737 | 0.95(0.75–1.20) | 0.663 |
| BMI < 18·5 and AGWG | 2·52(1·19–5·32) | 0.015 | 2·53(1·20–5·36) | 0.015 | 1.28(0.87–1.87) | 0.196 | 1.27(0.87–1.86) | 0.217 |
| BMI < 18·5 and IGWG | 2·43(1·40–4·21) | 0.002 | 2·46(1·41–4·28) | 0.002 | 1.68(1.34–2.12) | 0.000 | 1.67(1.32–2.10) | 0.000 |
| BMI < 18·5 and EGWG | 1·38(0·43–4·47) | 0.591 | 1·36(0·42–4·43) | 0.606 | 1.49(0.98–2.28) | 0.065 | 1.43(0.94–2.19) | 0.098 |
| BMI ≥ 25·0 and AGWG | 2·15(1·09–4·26) | 0.028 | 2·15(1·08–4·27) | 0.029 | 0.99(0.69–1.43) | 0.968 | 1.03(0.72–1.48) | 0.879 |
| BMI ≥ 25·0 and IGWG | 0·96(0·40–2·34) | 0.933 | 0·98(0·40–2·43) | 0.960 | 0.77(0.52–1.14) | 0.185 | 0.81(0.54–1.20) | 0.288 |
| BMI ≥ 25·0 and EGWG | 1·96(1·16–3·32) | 0.012 | 1·95(1·15–3·32) | 0.013 | 0.81(0.63–1.06) | 0.123 | 0.83(0.64–1.09) | 0.179 |

Values are calculated using GLMs of the Binomial or Poisson family with a log-link function, compared with reference value, normal BMI with AGWG, and presented as RR and 95% confidence intervals, and p value.

AGWG, adequate gestational weight gain; BMI, body mass index; CI, confidence intervals; EGWG, excessive gestational weight gain; IGWG, inadequate gestational weight gain; RR, risk ratio.

[a]Adjusted variables: Maternal age, maternal education, family wealth quintile and intervention.

proportion with EGWG was comparable. The review highlighted nutritional, behavioural, and clinical determinants as key drivers of IGWG. Compared with another study from India [31], our cohort had a higher prevalence of both IGWG and EGWG, and a lower prevalence of AGWG. That study reported a strong association between lower energy and fat intake and IGWG, underscoring the importance of dietary quality and quantity during pregnancy.

In our setting, pregnant women generally consumed less food than the amounts recommended by the Indian Council of Medical Research, which may be related to pregnancy-related symptoms such as loss of appetite, abdominal fullness, and bloating [31]. In addition, women may have eaten smaller portions and a less diverse diet because of household dietary norms, gendered food allocation, and constrained food availability. Together, these findings suggest that both inadequate and excessive GWG in this population are shaped by a complex interplay of individual symptoms, nutritional intake, and structural barriers to food access, and highlight the need for context-specific dietary counselling and social support interventions during pregnancy.

Several potential mechanisms may underlie these findings. Higher GWG is linked to increased maternal plasma volume expansion, greater fat deposition, and enhanced placental nutrient transfer, all of which support optimal fetal growth [32]. Conversely, IGWG can impair placental function and restrict fetal growth. Excessive GWG may expose the fetus to elevate levels of glucose and fatty acids, which can positively influence birth weight and other outcomes [33]. Inadequate GWG has also been associated with an increased risk of premature rupture of membranes, reproductive tract infections, and preterm birth. In contrast, EGWG is linked to conditions such as preeclampsia and edema, both of which are risk

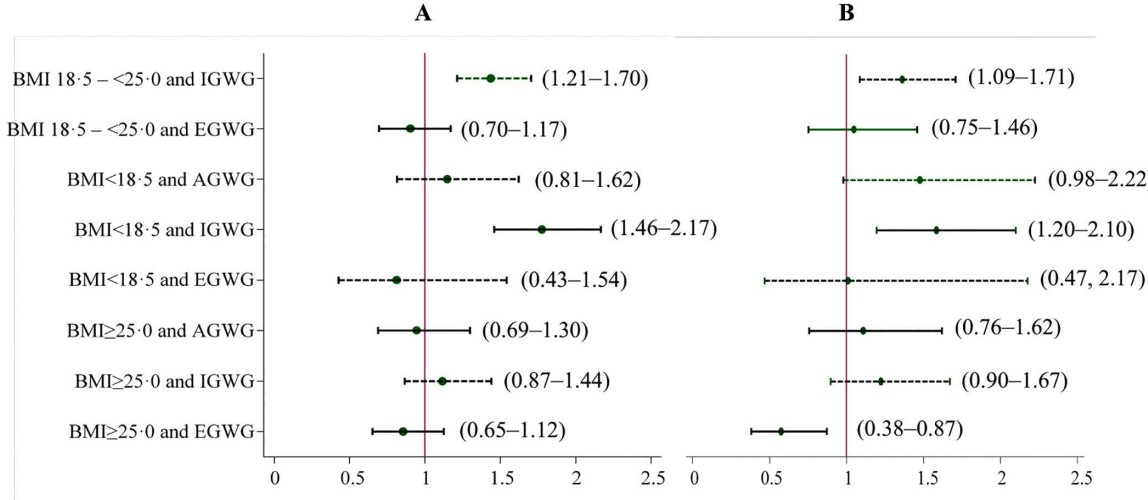

**Fig 3. GLMs of the Binomial or Poisson family with a log-link function was used to calculate RR after adjusting for maternal age, maternal education, family wealth quintile, and intervention effect.** Early pregnancy nutritional status was defined using the following cut-off pints: underweight (BMI < 18.5), normal weight (BMI 18.5–24.9), and overweight or obese (BMI ≥ 25). GWG was categorized as inadequate, adequate and excessive GWG according to the IOM cut-off points. Normal weight with AGWG were used as comparator to calculated RR and presented on X-axis with 95%CI. AGWG, adequate gestational weight gain; BMI, body mass index; CI, confidence intervals; EGWG, excessive gestational weight gain; GLM, generalized linear model; GWG, gestational weight gain; IGWG, inadequate gestational weight gain; IOM, Institute of medicine; RR, risk ratio.

factors for preterm delivery [34,35]. The study also highlighted a high prevalence of underweight (15%) and anaemia (22%) among women, with underweight women having a 30% higher risk of anaemia. Both anaemia and undernutrition are well-established risk factors for intrauterine growth restriction [36]. Notably, underweight women (BMI < 18.5 kg/m²) with EGWG faced an increased risk of fetal growth restriction and adverse birth outcomes.

The study has some limitations. Maternal weight was not recorded at birth to calculate the total weight gain during entire pregnancy. Several potential confounders were not measured, such as physical activity, and dietary intake which could have provided a more comprehensive understanding of factors influencing GWG and pregnancy outcomes. Additionally, while we screened for clinical intermediaries such as gestational diabetes, pregnancy-induced hypertension, and anemia, these were excluded from the final multivariable models because their inclusion did not alter the risk ratios or $\beta$-coefficients of the primary outcomes by the pre-specified threshold of 5%–10%. While this parsimonious approach focuses on the most statistically influential predictors, we acknowledge that these clinical conditions remain biologically relevant. Their exclusion might limit our ability to explore specific pathological pathways, though it ensures the stability and interpretability of our primary associations. Lastly, infant anthropometric outcomes were assessed within <14 days of birth, which may have introduced some variability in measurements due to postnatal weight and length changes during this period.

This study has several notable strengths. The prospective cohort design, with multiple antenatal weight measurements, allowed for trimester-specific assessment of GWG, providing detailed insights into its impact on pregnancy outcomes. The large sample size, combined with the inclusion of a significant proportion of underweight women, enabled robust comparisons across BMI categories. Gestational age was precisely determined using high-resolution ultrasonography.

## Conclusion

This study demonstrates that underweight women with inadequate gestational weight gain face the highest risk of adverse outcomes compared to normal BMI and adequate gestational weight gain.

**Table 4.** Association between early pregnancy BMI and GWG with wasting during second and third trimesters.

| Variables | Wasting (RR 95% CI) | | | |
|---|---|---|---|---|
| | Unadjusted | *P* value | Adjusted[a] | *P* value |
| *GWG from enrolment to 26 weeks of gestation (n = 3502)* | | | | |
| BMI 18·5 – <25·0 and AGWG | Ref | | Ref | |
| BMI 18·5 – <25·0 and IGWG | 1·11(0·89–1·38) | 0.354 | 1.05(0.84–1.31) | 0.660 |
| BMI 18·5 – <25·0 and EGWG | 0·73(0·51–1·03) | 0.077 | 0.74(0.52–1.05) | 0.089 |
| BMI < 18·5 and AGWG | 1·26(0·86–1·84) | 0.245 | 1.20(0.82–1.77) | 0.343 |
| BMI < 18·5 and IGWG | 1·25(0·93–1·67) | 0.134 | 1.19(0.89–1.60) | 0.234 |
| BMI < 18·5 and EGWG | 0·85(0·42–1·73) | 0.657 | 0.85(0.42–1.72) | 0.651 |
| BMI ≥ 25·0 and AGWG | 0·66(0·42–1·03) | 0.067 | 0.64(0.41–1.00) | 0.051 |
| BMI ≥ 25·0 and IGWG | 1·03(0·77–1·37) | 0.850 | 0.97(0.73–1.30) | 0.859 |
| BMI ≥ 25·0 and EGWG | 0·51(0·34–0·79) | 0.002 | 0.52(0.34–0.80) | 0.003 |
| *GWG from 27 weeks to the last visit of before delivery (n = 2976)* | | | | |
| BMI 18·5 – <25·0 and AGWG | Ref | | Ref | |
| BMI 18·5 – <25·0 and IGWG | 1·47(1·12–1·93) | 0.005 | 1.43(1.09–1.88) | 0.009 |
| BMI 18·5 – <25·0 and EGWG | 1·14(0·84–1·55) | 0.393 | 1.16(0.86–1.58) | 0.331 |
| BMI < 18·5 and AGWG | 1·40(0·84–2·33) | 0.197 | 1.37(0.83–2.28) | 0.221 |
| BMI < 18·5 and IGWG | 1·52(1·08–2·13) | 0.016 | 1.47(1.04–2.07) | 0.029 |
| BMI < 18·5 and EGWG | 1·33(0·71–2·51) | 0.375 | 1.35(0.71–2.55) | 0.359 |
| BMI ≥ 25·0 and AGWG | 0·85(0·50–1·44) | 0.552 | 0.85(0.50–1.44) | 0.539 |
| BMI ≥ 25·0 and IGWG | 0·94(0·59–1·52) | 0.814 | 0.92(0.57–1.49) | 0.731 |
| BMI ≥ 25·0 and EGWG | 0·91(0·64–1·28) | 0.574 | 0.91(0.65–1.29) | 0.606 |

Values are calculated using GLMs of the Binomial or Poisson family with a log-link function, compared with reference value, normal BMI with AGWG, and presented as RR and 95% confidence intervals, and p value.

AGWG, adequate gestational weight gain; BMI, body mass index; CI, confidence intervals; EGWG, excessive gestational weight gain; IGWG, inadequate gestational weight gain; RR, risk ratio.

[a]Adjusted variables: Maternal age, maternal education, family wealth quintile and intervention.

## Recommendation

Regular monitoring of gestational weight gain is essential, and timely energy and protein supplementation should be ensured for women who are underweight and/or experience inadequate weight gain during pregnancy.

## Supporting information

**S1 Table.** Association between early pregnancy body mass index and gestational weight gain with infant weight across pregnancy. Values were calculated using GLMs of the Gaussian family with an identity-link function, compared with reference value, normal BMI with AGWG, and presented as β-coefficient and 95% CIs. AGWG, adequate gestational weight gain; BMI, body mass index; CI, confidence interval; EGWG, excessive gestational weight gain; GLM, generalized linear model; IGWG, inadequate gestational weight gain. [1]Adjusted variables: maternal age, maternal education, family wealth quintile and intervention.
(DOCX)

**S2 Table. Association between early pregnancy body mass index and gestational weight gain with infant length across pregnancy.** Values were calculated using GLMs of the Gaussian family with an identity-link function, compared with reference value, normal BMI with AGWG, and presented as β-coefficient and 95% CIs. AGWG, adequate gestational

**A**

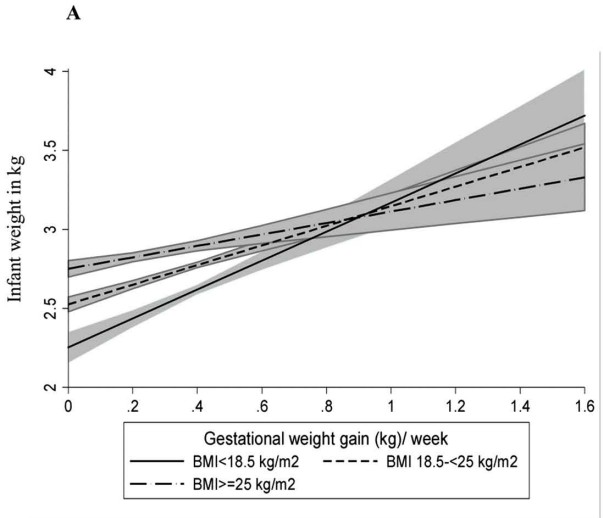

**B**

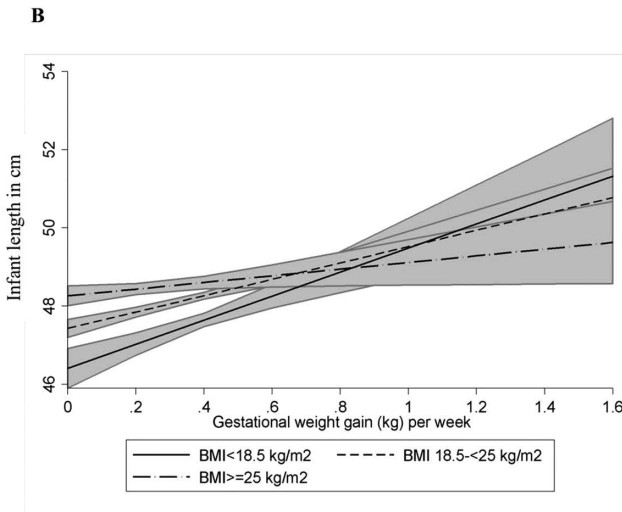

**Fig 4. The solid line indicates underweight (BMI<18·5 kg/m²), the dash line shows normal weight (BMI 18·5–<25 kg/m²) and the long dash dot line indicates overweight or obese (BMI≥25 kg/m²).** The bands represent 95% CIs from the linear regression model. BMI, body mass index; CI, confidence interval; GWG, gestational weight gain.

weight gain; BMI, body mass index; CI, confidence interval; EGWG, excessive gestational weight gain; GLM, generalized linear model; IGWG, inadequate gestational weight gain.[1]Adjusted variables: maternal age, maternal education, family wealth quintile and intervention.
(DOCX)

**S1 Fig. Directed Acyclic Graph, to shows the exposure, outcomes, confounders and mediators variables.** Exposure: Gestational weight gain & early pregnancy BMI; Outcome: Adverse pregnancy outcomes; Confounders changed risk ratio: Maternal age, maternal education, family wealth quintile and intervention; Potential confounders did not change risk ratio: Family type, anthropometry measurement duration, religion and parity; Potential mediator: Anaemia; BMI, body mass index.
(TIF)

**S2 Fig. Prevalence of small for gestational age, low birth weight, stunting and preterm by early pregnancy body mass index and total gestational weight gain.** AGWG, adequate gestational weight gain; BMI, body mass index; EGWG, excessive gestational weight gain; IGWG, inadequate gestational weight gain; LBW, low birth weight; SGA, small for gestational age.
(TIF)

**S3 Fig. Prevalence of small for gestational age, low birth weight, stunting and preterm by early pregnancy body mass index and gestational weight gain (from enrolment to 26 weeks of gestation).** AGWG, adequate gestational weight gain; BMI, body mass index; EGWG, excessive gestational weight gain; IGWG, inadequate gestational weight gain; LBW, low birth weight; SGA, small for gestational age.
(TIF)

**S4 Fig. Prevalence of small for gestational age, low birth weight, stunting and preterm by early pregnancy body mass index and gestational weight gain (from 27 weeks of gestation to last visit of before delivery). legend.** AGWG, adequate gestational weight gain; BMI, body mass index; EGWG, excessive gestational weight gain; IGWG, inadequate gestational weight gain; LBW, low birth weight; SGA, small for gestational age.
(TIF)

## Acknowledgments

We express our sincere gratitude to the participants and their families who participated in this study. We would also like to thank the data collection team for their contribution, and data management team for their digital data collection form development and data quality control.

## Author contributions

**Conceptualization:** Sunita Taneja, Nita Bhandari.

**Data curation:** Neeta Dhabhai.

**Formal analysis:** Saijuddin Shaikh, Ranadip Chowdhury.

**Methodology:** Ranadip Chowdhury.

**Supervision:** Neeta Dhabhai.

**Writing – original draft:** Saijuddin Shaikh, Ranadip Chowdhury.

**Writing – review & editing:** Sunita Taneja, Nita Bhandari.

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
