## [Decision Letter · Decision Letter 0]

15 Dec 2025

PGPH-D-25-02926

Impact of early pregnancy body mass index and gestational weight gain on birth outcomes: Findings from a pregnancy cohort in South Delhi, India

Dear Dr. Chowdhury,

Thank you for submitting your manuscript to PLOS Global Public Health. After careful consideration, we feel that it has merit but does not fully meet PLOS Global Public Health’s publication criteria as it currently stands. Therefore, we invite you to submit a revised version of the manuscript that addresses the points raised during the review process.

We look forward to receiving your revised manuscript.

Kind regards,

Nicola Hawley

Academic Editor

Journal Requirements:

1. Please clarify all sources of funding (financial or material support) for your study. List the grants (with grant number) or organizations (with url) that supported your study, including funding received from your institution.

2. State the initials, alongside each funding source, of each author to receive each grant.

3. State what role the funders took in the study. If the funders had no role in your study, please state: “The funders had no role in study design, data collection and analysis, decision to publish, or preparation of the manuscript.”

4. If any authors received a salary from any of your funders, please state which authors and which funders.

2. Please provide separate figure files in .tif or .eps format.

3. Please insert an Ethics Statement at the beginning of your Methods section, under a subheading 'Ethics Statement'. It must include:

1) The name(s) of the Institutional Review Board(s) or Ethics Committee(s)

2) The approval number(s), or a statement that approval was granted by the named board(s)

3) (for human participants/donors) - A statement that formal consent was obtained (must state whether verbal/written) OR the reason consent was not obtained (e.g. anonymity). NOTE: If child participants, the statement must declare that formal consent was obtained from the parent/guardian.

Additional Editor Comments:

As you will see from the reviews, both reviewers were enthusiastic about the manuscript but requested additional clarity in terms of methods and potential limitations. Please pay particular attention to making sure all of the possible limitations are described.

Reviewers' comments:

Reviewer's Responses to Questions

**Comments to the Author**

1. Does this manuscript meet PLOS Global Public Health’s publication criteria?

Reviewer #1: Yes

Reviewer #2: Yes

2. Has the statistical analysis been performed appropriately and rigorously?

Reviewer #1: Yes

Reviewer #2: I don't know

3. Have the authors made all data underlying the findings in their manuscript fully available (please refer to the Data Availability Statement at the start of the manuscript PDF file)?

Reviewer #1: Yes

Reviewer #2: No

4. Is the manuscript presented in an intelligible fashion and written in standard English?

Reviewer #1: Yes

Reviewer #2: No

Reviewer #1: The work is Excellence and the idea is novel However, the authors needs to address these minor comments. The authors need to review the abstract, methodology, ethical issues, discussion , conclusion and recommendation and references but add the limitation section which is not included in the manuscript.

Reviewer #2: This Study seems to be an extension of Previous published Study -WINGS.Data from that Study has been used and it has been declared by the Authors.Statistical Analyses has to be checked by Domain experts.Certain Tables and Figures mentioned in the text are not available for perusal.Few corrections in presentation and wordings are mentioned in my report which would be uploaded.

**Do you want your identity to be public for this peer review?** For information about this choice, including consent withdrawal, please see our Privacy Policy

Reviewer #1: **Yes:** Jonhas Masatu Malija

Reviewer #2: **Yes:** DR.CHANDRAKALA MARAN

---

## [Decision Letter · Decision Letter 1]

13 Jan 2026

PGPH-D-25-02926R1

Impact of early pregnancy body mass index and gestational weight gain on birth outcomes: Findings from a pregnancy cohort in South Delhi, India

Dear Dr. Chowdhury,

Thank you for submitting your manuscript to PLOS Global Public Health. After careful consideration, we feel that it has merit but does not fully meet PLOS Global Public Health’s publication criteria as it currently stands. Therefore, we invite you to submit a revised version of the manuscript that addresses the points raised during the review process.

Thank you for the careful attention to the initial reviewer comments. You will see that one of the reviewers raised some additional points for your consideration. Of those comments, I would appreciate your attention to the final one, which raises an important point about potential confounders. Please could you add a comment about that to the discussion?

We look forward to receiving your revised manuscript.

Kind regards,

Nicola Hawley

Academic Editor

Journal Requirements:

Reviewers' comments:

Reviewer's Responses to Questions

**Comments to the Author**

Reviewer #1: All comments have been addressed

Reviewer #2: (No Response)

publication criteria?

Reviewer #1: Yes

Reviewer #2: Yes

3. Has the statistical analysis been performed appropriately and rigorously?

Reviewer #1: Yes

Reviewer #2: I don't know

4. Have the authors made all data underlying the findings in their manuscript fully available (please refer to the Data Availability Statement at the start of the manuscript PDF file)?

Reviewer #1: Yes

Reviewer #2: No

5. Is the manuscript presented in an intelligible fashion and written in standard English?

Reviewer #1: Yes

Reviewer #2: Yes

Reviewer #1: The authors have addressed all comments related to ethical issues raised by reviewers. I don't have ongoing concern in my perspective.

Reviewer #2: Few corrections are to be made following which the Article can be accepted.I have attached the needed changes.

Thank you.

**Do you want your identity to be public for this peer review?** For information about this choice, including consent withdrawal, please see our Privacy Policy

Reviewer #1: **Yes:** Jonhas Masatu Malija

Reviewer #2: **Yes:** Dr.Chandrakala Maran

---

## [Editor Report · Decision Letter 2]

19 Jan 2026

Impact of early pregnancy body mass index and gestational weight gain on birth outcomes: Findings from a pregnancy cohort in South Delhi, India

PGPH-D-25-02926R2

Dear Dr. Chowdhury,

We are pleased to inform you that your manuscript 'Impact of early pregnancy body mass index and gestational weight gain on birth outcomes: Findings from a pregnancy cohort in South Delhi, India' has been provisionally accepted for publication in PLOS Global Public Health.

Best regards,

Nicola Hawley

Academic Editor